# REducing Colonoscopies in patients without significant bowEl DiseasE: the RECEDE Study - protocol for a prospective diagnostic accuracy study

Christopher Bradley [1], Siew Wan Hee,[2] Lazaros Andronis,[2] Krishna Persaud,[3] Mark A Hull,[4] John Todd,[1] Sian Taylor-Phillips [2], Steve Smith [1,5] Rachel Constable,[1] Norman Waugh,[2] Ramesh P Arasaradnam,[1,6] The RECEDE Study Group

For numbered affiliations see end of article.

**Correspondence to**
Mr Christopher Bradley; Christopher.Bradley@uhcw. nhs.uk

## ABSTRACT

**Introduction** Demand for colonoscopies and CT colonography (CTC) is exceeding capacity in National Health Service Trusts. In many patients colonoscopies and CTCs show no significant bowel disease (SBD). Faecal Immunochemical Testing (FIT) is being introduced to prioritise patients for colonoscopies but is insufficient to identify non-SBD patients meaning colonoscopy and CTC demand remains high. The REducing Colonoscopies in patients without significant bowEl DiseasE (RECEDE) study aims to test urine volatile organic compound (VOC) analysis alongside FIT to improve detection of SBD and to reduce the number of colonoscopies and CTCs.

**Methods and analysis** This is a multicentre, prospective diagnostic accuracy study evaluating whether stool FIT plus urine VOC compared with stool FIT alone improves detection of SBD in patients referred for colonoscopy or CTC due to persistent lower gastrointestinal symptoms. To ensure SBD is not missed, the dual test requires a high sensitivity, set at 97% with 95% CI width of 5%. Our assumption is that to achieve this sensitivity requires 200 participants with SBD. Further assuming 19% of all participants will have SBD and 55% of all participants will return both stool and urine samples we will recruit 1915 participants. The thresholds for FIT and VOC results diagnosing SBD have been pre-set. If either FIT or VOC exceeds the respective threshold, the participant will be classed as having suspected SBD. As an exploratory analysis we will be testing different thresholds. The reference comparator will be a complete colonoscopy or CTC. Secondary outcomes will look at optimising the FIT and VOC thresholds for SBD detection. An economic evaluation, using a denovo decision analytic model, will be carried out determine the costs, benefits and overall cost-effectiveness of FIT +VOC vs FIT followed by colonoscopy.

**Ethics and dissemination** Ethical approval was obtained by Liverpool Central Research Ethics Committee (20/ NW/0346).

**Trial registration number** RECEDE is registered on Clinicaltrials.gov NCT04516785 & ISRCTN14982373. This protocol was written and published before results of the trial were available.

## Strengths and limitations of this study

► Simple observational trial design that does not affect the standard care of participants.
► The study can be conducted entirely remotely.
► Reliance on patients collecting their samples at home and posting/bringing their samples back to the hospital may result in a higher proportion of missed samples as there is a greater chance of patients forgetting.
► Recruitment rate is dependent on COVID-19 19 as referral rates drop when COVID-19 19 cases are high.[4]

CT colonography (CTCs) within the National Health Service (NHS). Before the COVID-19 pandemic, around 300 000 patients were being referred annually to NHS trusts suspected of having colorectal cancer (CRC) and this number was rising.[1] Most patients referred with suspected CRC are offered an invasive colonoscopy or CTC examination, but only 30% of these participants have significant bowel disease (SBD).[2] SBD is defined as CRC, adenomatous polyps or inflammatory bowel disease (IBD).[3] The remaining 70% have a normal examination with 30% of those having functional conditions such as irritable bowel syndrome (IBS).

Set against this there will remain a capacity shortfall for colonoscopies and CTCs for the foreseeable future. This limits capacity of the NHS to extend CRC detection within the Bowel Cancer Screening Programme (BCSP) or to investigate those who present through the emergency department. The imbalance between capacity and demand, coupled with a lack of an accurate triage test makes it difficult for the NHS to stratify patients who present with bowel symptoms to those most at

## INTRODUCTION

There is currently a disparity between demand and available resources for colonoscopies and

risk of SBD. In the COVID-19 era, faecal immunochemical testing (FIT) is being introduced to triage patients who present with lower gastrointestinal (GI) symptoms.[4] However, almost all patients still receive a colonoscopy or alternative colonic imaging at some time and the optimum threshold for FIT has yet to be determined. The National Institute for Health and Care Excellence (NICE) have recommended $10\mu gHb/g$ faeces,[5] whereas the BCSP have set a cut-off of $120\mu gHb/g$ faeces.[6] Even at the lower threshold some SBD cases can be missed.[2 7 8] Previous data have shown that FIT alone (at a threshold of $7\mu gHb/g$ faeces) has a sensitivity of 80%, 53% and 86% for CRC, adenomatous polyps and IBD respectively.[2] A lower FIT threshold may reduce the number of false negatives but it will increase the number of unnecessary colonoscopies and CTCs in patients with no abnormalities (false positives). Moreover it will always miss cases of SBD where there is no bleeding, and a high proportion of malignant tumours never bleed.

Thus, this study will test urine volatile organic compound (VOC) analysis alongside FIT to improve detection of SBD. VOCs are produced by metabolic responses to inflammation in the presence of illness, therefore they are disease specific.[9–11] Previous work has identified key metabolites in inflammatory GI disease[12] and, on its own, VOC analysis has a sensitivity of 80%, 92% and 86% for CRC, adenomatous polyps and IBD, respectively.[13 14] When FIT and VOC are used in combination (FIT followed by VOC in the FIT negative), sensitivity of CRC improves from 80% to 97%, which is similar to the results of colonoscopy and CTC.[13 15] Existing studies have already demonstrated the utility of VOC analysis in detecting IBD, coeliac disease and bile acid diarrhoea while also being negative in functional conditions such as IBS.[16 17] Many patients with slightly raised FIT are found to have no SBD and we will examine the value of VOC in this group to see if a negative VOC can rule out SBD.

Therefore, the objective of this study is to investigate whether the combination of FIT and VOC analysis compared with FIT alone improves detection of SBD in patients who present with lower GI symptoms. This could be by both improving sensitivity in patients with FIT <10 and but also possibly by improving specificity in patients with slightly raised FIT but negative VOC. If correct, fewer patients will be referred for colonoscopies and CTCs, freeing up capacity and reducing NHS costs, while reducing disutility in patients who do not benefit from an invasive examination. Within the COVID-19 era, this has added importance. If the proposed method of dual testing proves effective then not only will hospital capacity be improved but also fewer patients will be required to undergo colonoscopies or CTCs; streamlining the patient pathway and reducing the risk of viral transmission if colonoscopy generates aerosols.

## METHODS AND ANALYSIS
### Study design and procedures
This is a multicentre (UK only), prospective diagnostic accuracy study, sponsored by the University Hospitals Coventry & Warwickshire (UHCW) to compare dual testing of stool FIT plus urine VOC markers compared with stool FIT alone in the detection of SBD. Participants will be recruited when referred to secondary care for investigation of lower bowel symptoms, after informed consent is received. Inclusion criteria will be: patients referred non-urgently or urgently (fulfilling the national criteria for referral—NICE NG12)[18] with lower GI symptoms and determined by the overseeing clinician to require colonoscopy or CTC; minimum age of 18; able to provide informed consent, and have the ability to return both stool and urine samples. Exclusion criteria are those who are pregnant. Participants will be recruited over the phone or face-to-face during a clinic visit which precedes their colonoscopy or CTC. Participants will be asked to provide a stool sample and urine sample at least 24 hours before commencing their bowel cleansing medication. Stool samples will be collected at home by the participants using FIT collection kits and posted directly to the analysis site. If the participant has already completed a FIT during their referral pathway, we will use this data if the time between the FIT and their colonoscopy is less than 4 weeks, otherwise we will request that the participant to provide another FIT. Urine samples will be returned to hospitals and frozen at −80°C pending future analysis. Urine samples can either be collected during a clinic visit that precedes their colonoscopy or CTC, or collected at the home of the participant and brought back by the participant. We will aim for the time between the urine sample being produced and the urine sample being frozen at −80°C to be less than 4 hours, otherwise we will ask participants to freeze their urine samples in their home freezer. For the purposes of the economic analysis, a subset of participants (20% of total) undergoing colonoscopy or CTC will also be asked to complete questionnaires at five time points surrounding their examination. This will help determine the disutility and costs associated with undergoing colonoscopy or CTC, which will be used as inputs in the economic model (see section Economic Evaluation below). Figure 1 shows a participant flow diagram once enrolled in the REducing Colonoscopies in patients without significant bowEl DiseasE (RECEDE) study.

### Sample size calculation
To replace colonoscopy—the current gold standard for diagnosing bowel conditions—and/or CTC, FIT and VOC analysis requires a very high sensitivity to avoid false negatives. High specificity is also desirable, to reduce false positives. Based on our previous study, the sensitivity of FIT and VOC analysis to detect CRC was 97%, comparable to colonoscopy. Therefore, the sample size was determined to achieve a sensitivity of at least 97% with a 95% CI width of 5% (Zhou-Li method[19] as described in box 1).

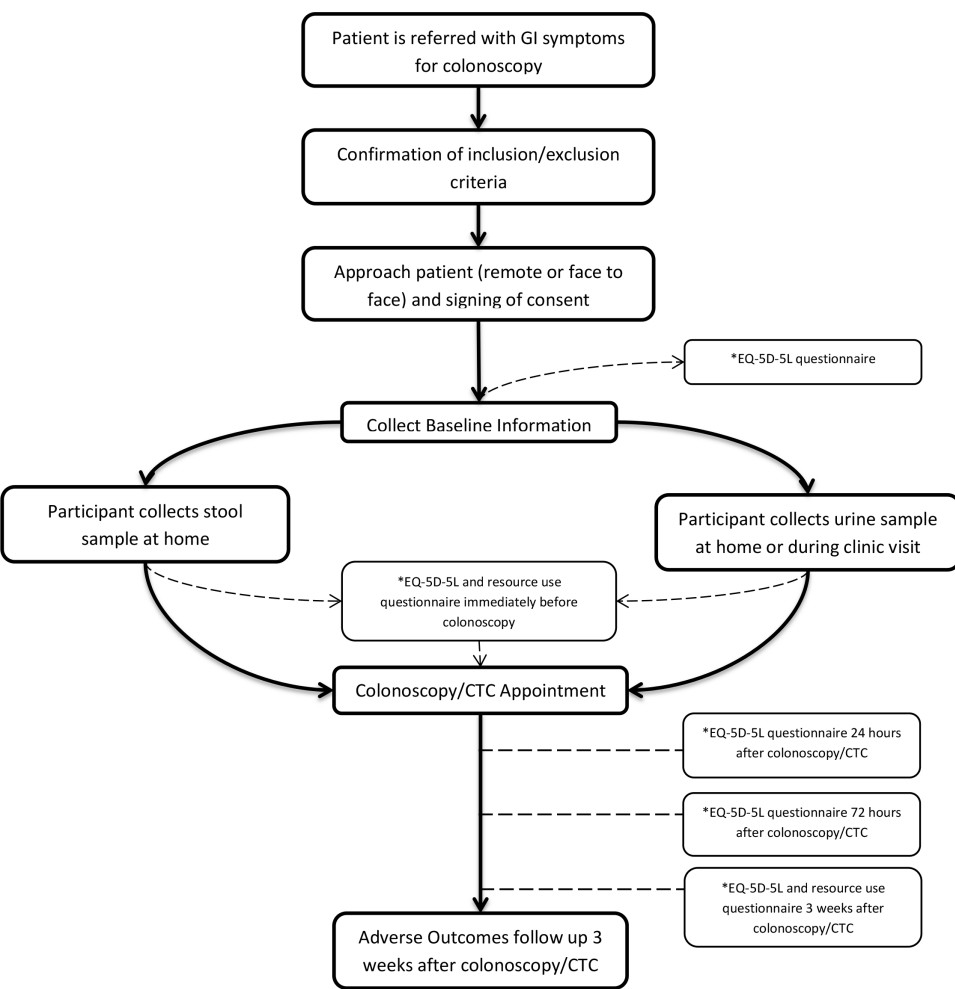

**Figure 1** Participant flow diagram for RECEDE. *Indicates that the data is only being collected in 20% of participants. CTC, CT colonography; EQ-5D-5L, EuroQol-5 Dimension-5 Level; GI, gastrointestinal; RECEDE, REducing Colonoscopies in patients without significant bowEl DiseasE.

The required number of participants with SBD to achieve this sensitivity is 200. Figure 2 shows CIs for different observed sensitivity cases of SBD (150, 200, 250). Figure 3 shows the widths of the CI limits are narrower and wider when the observed sensitivity is above or below 97%,[13] respectively. This sample size is also robust to negative predictive value (NPV), an outcome that is of interest as a high NPV suggests that participants without SBD would avoid having colonoscopy examination. Assuming the prevalence of SBD in the study population is 19%, then 1053 participants are required. Based on our previous study,[13] the return rate of both stool and urine samples that are eligible for analysis is about 55% resulting in a total required sample size of 1915 participants across multiple UK sites within a 24-month period.

### Stool sample analysis

Stool samples will be analysed for traces of haemoglobin in the faeces using a HM-JACKarc. Participants samples will be defined as having suspected SBD if the value is >10 $\mu$ gHb/g faeces. Any previous FIT results collected to triage patients can be sourced from patient records if the test is <4 weeks before their colonoscopy or CTC.

### Urine VOC analysis

Participants will be asked to collect duplicate urine samples in universal sterilin pots. Urine samples will be transported to the analysis site using dry ice to maintain the cold chain. All urine samples will be analysed using the 'electronic nose'[16] technique. Briefly, this technique enables separation of VOCs between disease groups based on chemical fingerprint pattern rather than specific

---

**Box 1    Sample size determination based on CI method**

The Zhou Li CI is $\left(\frac{e^{LL}}{1+e^{LL}}, \frac{e^{UL}}{1+e^{UL}}\right)\left(\frac{e^{LL}}{1+e^{LL}}, \frac{e^{UL}}{1+e^{UL}}\right)$ where

$LL = ln\left(\frac{\bar{S}e}{1-\hat{S}e}\right) - \frac{1}{\sqrt{n\hat{S}e(1-\hat{S}e)}} g^{-1}(z_{1-\alpha/2})$

$UL = ln\left(\frac{\bar{S}e}{1-\hat{S}e}\right) - \frac{1}{\sqrt{n\hat{S}e(1-\hat{S}e)}} g^{-1}(z_{\alpha/2}),$

$g^{-1}(x) = -\sqrt{n}\left(\frac{6}{\hat{\gamma}}\right)\left\{\left[1 - \frac{\hat{\gamma}}{2}\left(\frac{x}{\sqrt{n}} - \frac{\hat{\gamma}}{6n}\right)\right]^{1/3} - 1\right\}$

and $\hat{\gamma} = \frac{1-2\bar{S}e}{\hat{S}e(1-\hat{S}e)}$

---

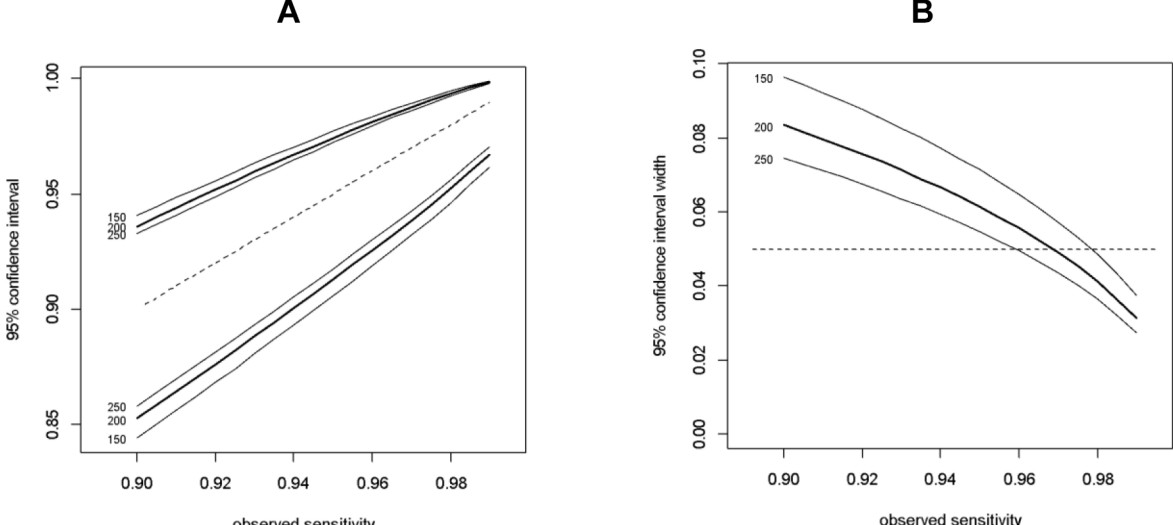

**A**

**B**

**Figure 2** CIs for different estimates of sensitivity for 150, 200 and 250 cases of SBD. SBD, significant bowel disease.

chemical analysis. This allows screening of populations rapidly using chemometric data analysis (information extracted from chemical systems) and receiver operator curves (ROC) to distinguish one population from another, with ROCs being used to measure the appropriate thresholds to be set to distinguish samples as having SBD. A sample which has a ROC with an area under the curve of 0.63 or greater will be defined as having suspected SBD.

A subset of 100 urine samples will also be analysed using a gas-chromatography mass spectroscopy system. This technique separates complex mixtures of chemicals based on their interaction with a retentive layer, resulting in chemicals eluding out of the gas-chromatography at different times. These individual chemicals are then ionised and the mass of the resultant fragments measured. The analysis will be split into two stages. First, a subset of SBD positive and SBD negative samples will be screened to identify up to 8 VOCs that are prominent in SBD positive samples. Once identified, a

headspace gas chromatography/mass spectrometry method will be developed to detect, separate and quantify the target VOCs selected in stage one within the remaining samples. This method will highlight whether specific VOCs present more frequently in SBD positive samples.

### Adoption of study due to COVID-19 pandemic: remote delivery

In order to continue recruiting through the COVID-19 pandemic, an innovative study design was adopted whereby participant involvement shifted to an entirely remote approach. This allowed the study to continue recruiting through the peak waves of COVID-19 when many other studies were forced to temporarily pause.[20]

### Reference test

The reference test is the final report from the colonoscopy or CTC examination and confirmatory histology report. Biopsies are required as per national guidance[5] in

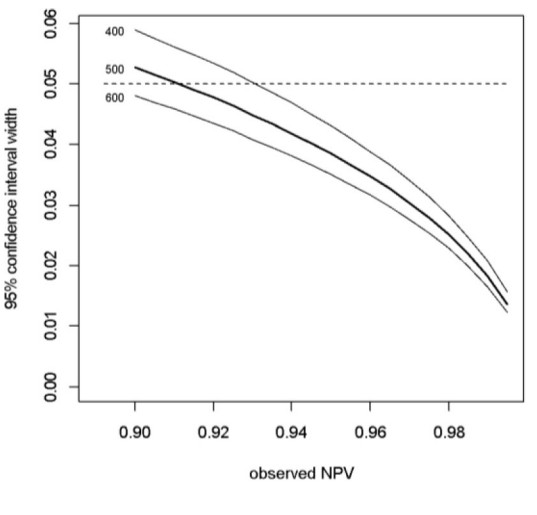

**A**

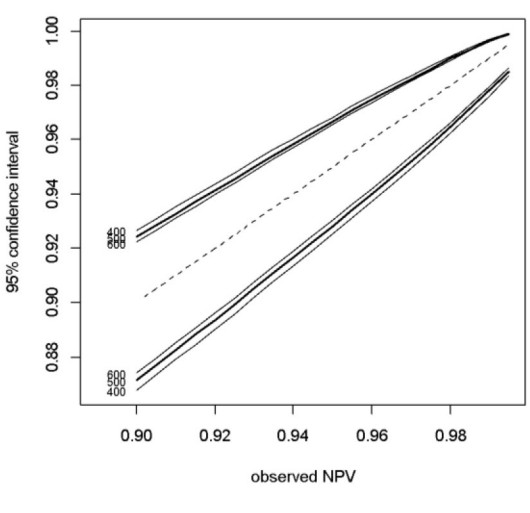

**B**

**Figure 3** CIs for different estimates of NPV for 400 500 and 600 negative cases of SBD from 1000 samples. NPV, negative predictive value; SBD, significant bowel disease.

the investigation of participants with lower GI symptoms even in the absence of macroscopic abnormalities. Those who perform the reference test will have no knowledge of the results of the index tests and vice versa.

## Data management

Data from the study will be stored on an online validated Good Clinical Practice (GCP) compliant electronic data capture (EDC) system. Individual user access will be provided only to members of the research team who require it for their role. No personal data will be uploaded to the EDC. Participant identification codes will be used to ensure pseudoanonymisation. Following the resolution of queries, the database will be locked and exported to the trial statistician for analysis. Access to the final trial dataset will only be made available to those who require it for the analysis.

## Primary outcome analysis

The primary outcome is diagnostic accuracy (sensitivity, specificity, NPV and positive predictive value (PPV)) of the dual index test FIT and VOC analysis. The 'or' rule will be utilised for the primary analysis whereby a FIT result of >10 µgHb/g faeces OR a VOC result with predicted probability >0.63 implies that the participant is suspected of having SBD and needs to be referred for colonoscopy or CTC. We will report 2×2 tables of test accuracy of the dual test by the reference standard (final report from the colonoscopy examination and confirmatory histology report of SBD which is either colon cancer, colorectal adenomas or IBD), alongside summary measures of sensitivity, specificity, NPV and PPV, and their 95% CIs. These test accuracy estimates will be used in the economic model.

We will analyse the combined FIT+VOC test results as both parallel (results from both tests are interpreted in combination) and serial testing (VOC analysis only performed if FIT is negative). The 'or' rule will be considered in the interpretation of both parallel and serial tests where the diagnosis is positive when either test is positive, and negative when both results are negative. In the serial test, if the FIT is positive, then the diagnosis is positive. If FIT is negative we will use the result from VOC analysis and if that is positive then the diagnosis is positive. Mathematically, both methods should give the same diagnostic results but will make a difference in the economic model.

## Secondary outcome analyses

Secondary outcomes include: ROC of FIT plus VOC to develop an optimum thresholds for SBD detection; sensitivity, specificity, PPV and NPV estimates and their corresponding 95% CI for each individual condition of SBD (CRC, colorectal adenomas and IBD) by each individual test and combination rules; compliance rates (return of samples); the types of SBD and microscopic colitis, and the number of cases missed by CTC, FIT and VOC analyses; potential number of avoidable colonoscopies and CTCs in patients without SBD; total NHS and personal social services costs and total quality of adjusted life years (QALY) associated with each option. To find the optimal thresholds of FIT and

VOC we will first plot the ROC curves of both tests individually to find the individual optimal thresholds (the one that gives the greatest specificity while achieving 97% sensitivity). Second, we will plot combinations of the two tests in the ROC space and similarly, choose the combination of thresholds which has greatest specificity while achieving the required 97% sensitivity to avoid colonoscopy. The combined thresholds of FIT and VOC in the ROC space are calculated by combining every unit of FIT (from 10 to 120µgHb/g faeces) with every unit of the predicted probability of VOC (from 0.63 to 1). Both the 'or' and 'and' rules will be explored when combining the FIT and VOC results to define a case as either SBD positive or SBD negative. If both tests are in discordant then the diagnosis is negative. For each of the thresholds that we derive from the ROC curves, we will also report the NPV estimates. This will produce an overestimate of test accuracy, which we will use as a sensitivity analysis in the economic model, as a more optimistic assumption to demonstrate the range of uncertainty.

## Economic evaluation

An economic analysis will be carried out to determine the costs, benefits and overall cost-effectiveness of FIT +VOC versus FIT alone in selecting patients for colonoscopy or CTC among those with GI symptoms referred for investigation. The analysis will be carried out primarily from the perspective of NHS and it will employ a de novo analytic model built as part of this study. In line with recent literature,[7] it is envisaged that the model will employ a multipart structure, consisting of a decision tree to evaluate short-term cost and consequences accruing at the diagnosis stage followed by a state-transition model to capture the long-term (lifetime) outcomes associated with the diagnosed condition.

Calculations will consider key costs and consequences, including the costs and disutility associated with having a colonoscopy. Key model input will be drawn from the RECEDE study, including parameters related to the diagnostic characteristics of the compared options, estimates of the rate of adverse outcomes associated with colonoscopies (eg, bowel perforation and bleeding) and CTCs, as well as primary and secondary care resource use and preference-based health-related quality of life (utility) values associated with colonoscopy. The latter information will be needed to reflect the fact that undergoing a colonoscopy, including the bowel preparation and the procedure itself is an unpleasant activity which is likely to result in a temporary decrease in quality of life. Estimates of changes in quality of life will be obtained by administering a widely used and recommended generic questionnaire (EuroQol-5 Dimension-5 Level[21]) to a sample of 370 participants (20% of the total study sample) scheduled to undergo colonoscopy or CTC at (1) baseline (ie, when participant consent is received) (2) immediately prior to colonoscopy once the participant has fully completed bowel preparation, (3) 24 hours after the colonoscopy examination, (4) 72 hours post colonoscopy and (5) 3 weeks post colonoscopy. Use of resources, including NHS

care, out of pocket payments and loss of income the participants may occur as a result of their colonoscopy will also be collected through patient questionnaires administered to the same sample of participants at 3 weeks post colonoscopy or CTC.

In line with recommendations, results will be presented in terms of total costs per additional QALY associated with FIT +VOC compared with FIT alone.[22] Deterministic and probabilistic sensitivity analyses will be undertaken to explore the robustness of the obtained results to sample variability and plausible variations in key assumptions and employed analytical methods.[23 24] The model will also form the basis for conducting value of information analysis, which will quantify the total expected cost due to the remaining uncertainty around the decision problem[25 26]

### Peer review and patient and public involvement

The RECEDE study has been peer reviewed in the NIHR HSDR process by external reviewers and the commissioning board. This protocol has also been reviewed externally by a Trial Steering Committee and internally by the Trial Management Group. The study was also reviewed by members of the GUT club (survivors of GI cancer) and the patient and public involvement in research (PPI) at the sponsor site. Both groups were supportive of the aims of RECEDE and agreed further evidence is required for the utility of FIT and VOC analysis in diagnosing SBD.

### ETHICS AND DISSEMINATION

Ethical approval was provided by Liverpool Central Research Ethics Committee (20/NW/0346). The study will be conducted in compliance with the principles of the International Conference on Harmonisation (ICH) GCP guidelines and in accordance with all applicable regulatory guidance. The study will comply with the current Data Protection regulations and regular checks and monitoring will be undertaken by the study manger to ensure compliance.

Results from RECEDE will naturally be of profound interest to clinicians worldwide who investigate lower GI symptoms. Results can also be used by NICE to inform a revised pathway for managing participants with lower GI symptoms—to categorise high risk, that is, those with SBD versus low risk (without SBD). Those with very low risk of SBD could potentially be managed in primary care without need to refer to secondary care. RECEDE will lead to high level presentations at GI and Oncological meetings both nationally and internationally. It will also result in high quality open access manuscripts. A range of dissemination products will include annual reports, national publications, press releases through UHCW Communications Department, participant safety collaborations, presentation and talks as well as videos will ensure that all audiences can be updated. Dissemination of results to participants will be led by our PPI coapplicant and facilitated through GUTs UK which is the partner charity of the British Society of Gastroenterology. We will also engage with key stakeholders including PPI groups, local specialised colorectal Clinical Research Group, as well as Cancer Alliance groups

**Author affiliations**
¹Research & Development, University Hospitals Coventry and Warwickshire NHS Trust, Coventry, UK
²Warwick Clinical Trials Unit, University of Warwick, Coventry, UK
³Department of Chemical Engineering, The University of Manchester, Manchester, UK
⁴School of Medicine, University of Leeds, Leeds, UK
⁵Bowel Cancer Screening Hub, NW Bowel Cancer Screening Hub, Hospital of St. Cross, Rugby, UK
⁶Warwick Medical School, University of Warwick, Coventry, UK

**Collaborators** The Recede study group includes : Dr Sami Hoque, Mr Chris Smart, Dr Zia Mazhar, Dr Ioannis Koumoutsos, Dr Ravi Madhotra, Prof Jimmy Limdi, Mrs Rachel Walker, Prof Dean Harris, Dr Jervoise Andreyev, Dr Babu Sathish, Dr Said Din, Mrs Helena Cox, Prof Stephen Lewis.

**Contributors** (1) Substantial contribution to design of work. (2) Drafting and revising the intellectual content. (3) Final approval of version to be published. (4) Agreement to be accountable for works published. CB (1, 2, 3, 4), SWH (1, 2, 3, 4), LA (1, 2, 3, 4), KP (1, 2, 3, 4), MAH (1, 2, 3, 4), JT (1, 2, 3, 4) ST-P (1, 2, 3, 4), SS (1, 2, 3, 4), NW (1, 2, 3, 4), RC (1, 2, 3, 4), RA (1, 2, 3, 4).

**Funding** This study is funded by the National Institute for Health Research (NIHR) (Health Service and Delivery Research (project reference 127800)).

**Disclaimer** The views expressed are those of the authors and not necessarily those of the NIHR or the Department of Health and Social Care. In memory of John Todd, deceased 17 April 2021

**Competing interests** None declared.

**Patient and public involvement** Patients and/or the public were involved in the design, or conduct, or reporting, or dissemination plans of this research. Refer to the Methods section for further details.

**Patient consent for publication** Not applicable.

**Provenance and peer review** Not commissioned; externally peer reviewed.

**ORCID iDs**
Christopher Bradley http://orcid.org/0000-0002-6905-1768
Sian Taylor-Phillips http://orcid.org/0000-0002-1841-4346
Steve Smith http://orcid.org/0000-0002-3468-7483

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
