## [Reviewer comments · BMJ Open]

ARTICLE DETAILS

TITLE (PROVISIONAL)	REducing Colonoscopies in patients without significant bowEl Disease – The RECEDE Study - Protocol for a Prospective Diagnostic Accuracy Study.
AUTHORS	Bradley, Christopher; Hee, Siew Wan; Andronis, Lazaro; Persaud, Krishna; Hull, Mark A.; Todd, John; Taylor-Phillips, Sian; Smith, Steve; Waugh, Norman; Constable, Rachel; Arasaradnam, Ramesh

VERSION 1 – REVIEW

REVIEWER	Kozarek, Richard Virginia Mason Medical Center, Digestive Disease Institute
REVIEW RETURNED	05-Nov-2021

GENERAL COMMENTS	The current study has significant implications for the performance of diagnostic colonoscopy and CTC in Great Britain and worldwide as well as significant cost savings implications if the combination of VOC (portable electronic nose) and FIT-diagnosed individuals with and without significant bowel disease (SBD) is comparable to invasive procedures. 1. Yearly FIT tests are used variably throughout the world in Asymptomatic patients to check for elevated stool hemoglobin and need for colonoscopy. The investigators are studying an enriched pool of Symptomatic patients from the Emergency department and clinical practices. As such, findings in this populations may not be generalizable to a larger and asymptomatic patient group.2. The reviewer has an issue with the definition of SBD: cancer, adenomatous polyps, and inflammatory bowel disease (IBD).a. Are you defining microscopic/lymphocytic/collagenous colitis as IBD?b. Are cecal angiomata in a patient with iron deficiency anemia not significant? Appendiceal fecaliths or carcinoids? A post-diverticulitis stricture? The reviewer can think of a dozen additional significant diagnoses.3. The study fails to account for lab and clinical data for which the patient has been sent for colonoscopy: meds, nocturnal diarrhea, fecal incontinence, weight loss, iron deficiency, anemia...4. You are using colonoscopy as your gold standard. What is the documented rate of complete colonoscopy (to include ileoscopy?) in GB? What is the gold standard for an incomplete exam or a poorly prepped colon?5. The EQ-5D-5L questionnaire that is to be given multiple times to a subset of patients should have been included in this submission as the reviewer is uncertain that it captures patient relief or reassurance from a negative CTC or colonoscopy. It certainly captures the discomfort from rigid bowel prep and invasive procedures.
--

REVIEWER	Derakhshan, Mohammad University of Glasgow
REVIEW RETURNED	17-Nov-2021

GENERAL COMMENTS	The rising load of colonoscopy and CTC for the detection of colorectal malignancies and other significant bowel diseases is an undeniable issue not only for UK NHS but also for any other screening authority around the world. The recent COVID crisis has also heightened the problem significantly. Dr Bradley and colleagues know the NHS bottleneck very well, so trying to reduce the demand by recommending a new screening modality to prioritise symptomatic patients for colonoscopy and CTC. The protocol is well designed, and all aspects of the study have been clearly described. Here I have listed a few comments to clarify some points which are not clear to me or just a recommendation to expand the analysis of the ongoing study.  1. According to ref 3 only 30% of those participants referred to colonoscopy/CTC have SBD, but authors have used 19% for sample size calculation, please clarify. 2. Authors have correctly highlighted the insufficient sensitivity and specificity of FIT and the huge discrepancy between NICE and BCSP-recommended cut-offs. While this is a correct argument, the main shortfall of the FIT is that the test relies on bleeding as a sign of SBD, but a significant proportion of malignant tumours never bleed. This fact has been mentioned in the protocol but needs to be brought to attention explicitly. 3. Inclusion criteria: Authors have decided to recruit all patients (symptomatic) referred routinely or urgently to secondary care for colonoscopy/CTC. While this is a good reflection of real-world practice in NHS, we know the chance of finding a significant bowel pathology is different among two groups of patients. Therefore, I recommend having subgroups analyses when comparing the diagnostic test accuracy.
--

VERSION 1 – AUTHOR RESPONSE

Reviewer: 1

Dr. Richard Kozarek, Virginia Mason Medical Center Comments to the Author:

The current study has significant implications for the performance of diagnostic colonoscopy and CTC in Great Britain and worldwide as well as significant cost savings implications if the combination of VOC (portable electronic nose) and FIT-diagnosed individuals with and without significant bowel disease (SBD) is comparable to invasive procedures.

- 1. Yearly FIT tests are used variably throughout the world in asymptomatic patients to check for elevated stool haemoglobin and need for colonoscopy. The investigators are studying an enriched pool of Symptomatic patients from the Emergency department and clinical practices. As such, findings in this population may not be generalizable to a larger and asymptomatic patient group.**

We are not investigating asymptomatic patients and thus our results are not expected to be directly translatable to those without symptoms i.e. screening population. However the concept of using dual testing to improve diagnostic accuracy will be valuable to be evaluated in the screening population. As outlined, our population are those who present with symptoms and therefore it would be at this point that the FIT + VOC diagnostic tool is being proposed.

- 2. The reviewer has an issue with the definition of SBD: cancer, adenomatous polyps, and inflammatory bowel disease (IBD).**

Are you defining microscopic/lymphocytic/collagenous colitis as IBD?

No, inflammatory bowel disease will be defined as per the international classification of diseases. Originally we had included microscopic colitis within the SBD definition criteria – though separate to IBD. Due to COVID, we began to see an increasing numbers of patients being referred for CT scans/CTCs in an attempt to clear the backlog of patients on the waiting list. As CTC has comparable accuracy to colonoscopy, we wanted to capture these patients within RECODE. However CTC cannot detect microscopic colitis and hence we removed this disease from the SBD definition. Microscopic colitis prevalence is low (<0.5% of SBD cases) and so we didn't feel this would impact our outcomes significantly. We adjusted the samples size accordingly (from 1819 to 1915) so that the power calculation would not be affected.

Are cecal angiomata in a patient with iron deficiency anemia not significant? Appendiceal fecaliths or carcinoids? A post-diverticulitis stricture? The reviewer can think of a dozen additional significant diagnoses.

Indeed, there are many other conditions that could be deemed significant but may not necessarily warrant a colonoscopy. The aim of this study was to triage using dual markers those that we have defined as might have significant bowel disease that would warrant a colonoscopy. The conditions also have to be relatively common to our population. For angiomata that are bleeding, the FIT test will be raised which will trigger a colonoscopy. Post diverticulitis stricture can be imaged radiologically and would not warrant a colonoscopy.

- 3. The study fails to account for lab and clinical data for which the patient has been sent for colonoscopy: meds, nocturnal diarrhoea, faecal incontinence, weight loss, iron deficiency, anaemia.**

The referral criteria for our patients are based on NICE NG12 and DG30 criteria which in effect takes into account those with high and low risk of developing colorectal cancer. Please see links to NICE guidance now referenced in manuscript on line 87.

- 4. You are using colonoscopy as your gold standard. What is the documented rate of complete colonoscopy (to include ileoscopy?) in GB? What is the gold standard for an incomplete exam or a poorly prepped colon?**

Colonoscopy is the accepted gold standard for lower gastrointestinal investigations. The UK national audit suggests 85% completion rate. For those that are incomplete, either a repeat examination is scheduled or CT colon with contrast is organised. Our primary outcome is based on complete colonic examination.

- 5. The EQ-5D-5L questionnaire that is to be given multiple times to a subset of patients should have been included in this submission as the reviewer is uncertain that it captures patient relief or reassurance from a negative CTC or**

colonoscopy. It certainly captures the discomfort from rigid bowel prep and invasive procedures.

The EQ-5D-5L does not capture this information. It captures patient's quality of life to show a dip and subsequent recovery in QoL around a colonoscopy examination period. This will feed into the economic model. A separate study is being considered that will compare patient acceptability of a diagnosis from colonoscopy, CTC and FIT + VOC. However this requires an accurate quantification of the sensitivity and specificity of FIT + VOC which will be provided by RECEDE.

Reviewer: 2

Dr. Mohammad Derakhshan, University of Glasgow Comments to the Author:

The rising load of colonoscopy and CTC for the detection of colorectal malignancies and other significant bowel diseases is an undeniable issue not only for UK NHS but also for any other screening authority around the world. The recent COVID crisis has also heightened the problem significantly. Dr Bradley and colleagues know the NHS bottleneck very well, so trying to reduce the demand by recommending a new screening modality to prioritise symptomatic patients for colonoscopy and CTC.

The protocol is well designed, and all aspects of the study have been clearly described. Here I have listed a few comments to clarify some points which are not clear to me or just a recommendation to expand the analysis of the ongoing study.

- 1. According to ref 3 only 30% of those participants referred to colonoscopy/CTC have SBD, but authors have used 19% for sample size calculation, please clarify.**

The sample size calculation is based on an underestimate for prevalence of SBD to ensure we have enough cases of SBD when the study is completed. Originally we had assumed a 20% prevalence rate, however this was reduced to 19% when we removed microscopic colitis from the SBD definition.

2. **Authors have correctly highlighted the insufficient sensitivity and specificity of FIT and the huge discrepancy between NICE and BCSP-recommended cut-offs. While this is a correct argument, the main shortfall of the FIT is that the test relies on bleeding as a sign of SBD, but a significant proportion of malignant tumours never bleed. This fact has been mentioned in the protocol but needs to be brought to attention explicitly.**

This has been added into line 61 of the protocol. The reviewer has highlighted a shortfall with relying on FIT testing as a single test hence the rationale for dual testing with urinary volatile organic compounds.

3. **Inclusion criteria: Authors have decided to recruit all patients (symptomatic) referred routinely or urgently to secondary care for colonoscopy/CTC. While this is a good reflection of real-world practice in NHS, we know the chance of finding significant bowel pathology is different among two groups of patients. Therefore, I recommend having subgroups analyses when comparing the diagnostic test accuracy.**

The referral criteria for RECEDE are based on fulfilment of NG12 or DG30 NICE criteria which is for referrals categorised as urgent in those suspected with bowel cancer (now referenced on line 87). Routine referrals are usually from secondary care where bowel cancer is not suspected or those undergoing screening for colorectal cancer. These account for a small proportion of colonoscopy referrals and outwit our inclusion criteria. Hence, we are not able to distinguish between urgent and routine referrals. We have removed the word 'routine' from the manuscript and apologise for the confusion.

VERSION 2 – REVIEW

REVIEWER	Kozarek, Richard Virginia Mason Medical Center, Digestive Disease Institute
REVIEW RETURNED	15-Dec-2021

GENERAL COMMENTS	1. The addition of the online NICE 2015 guideline referenced in the manuscript on line 87 is appropriate and improves better understanding of the study. Note, "Cancer" is misspelled in reference #18, and there has been a 2020 update to this GL, entitled "Clinical guide for triaging patients with lower GI symptoms (for colorectal cancer)." 2. You state that the UK National Audit suggests an 85% colonoscopy completion rate, but that statistically your primary outcome is based upon complete examination. As a practitioner and clinical researcher, I find that incomplete colonoscopies, either from inadequate bowel preps (recommend recording the Boston Bowel Prep Scores), patient discomfort, or CR instability during the procedure approximates 7–8%. Less than half of those patients are willing or able to undergo a follow-up colonoscopy or CTC within a year. This suggests to me that a 15% incomplete colonoscopy may require larger patient numbers unless the authors have information that this 15% of patients all undergo a timely repeat colonoscopy or CTC in Britain.
---

REVIEWER	Derakhshan, Mohammad University of Glasgow
REVIEW RETURNED	08-Dec-2021
GENERAL COMMENTS	I am satisfied with the changes and modifications in the revised protocol, no more changes are required.

VERSION 2 – AUTHOR RESPONSE

1. Thank you for pointing out the spelling error. This has now been corrected in the reference. The 2015 reference that we have provided is linked to the NICE website which is continually updated to match the latest guidance. The last update of this web page was December 2021.

2. Our primary aim is complete colonic examination to identify significant bowel disease (SBD) which is how the study is powered. Hence only those that achieve this are included irrespective of variation in colonoscopy completion rates in individual recruitment centers. We require 200 cases of SBD. We assume that 19% of patients who have a complete colonoscopy will have SBD so we require 1053 participants who have a complete colonoscopy. We further assume that 55% of people will return both of their stool and urine samples leading us to an overall target of 1915 to achieve 200 cases of SBD.

The only change in the attached revised manuscript is the spelling error in reference 18.